# Features of the Preparation and Luminescence of Langmuir-Blodgett Films Based on the Tb(III) Complex with 3-Methyl-1-phenyl-4-stearoylpyrazol-5-one and 2,2′-Bipyridine

**DOI:** 10.3390/ma15031127

**Published:** 2022-01-31

**Authors:** Julia Devterova, Kirill Kirillov, Anton Nikolaev, Michail Sokolov, Victor Shul’gin, Alexey Gusev, Victor Panyushkin, Wolfgang Linert

**Affiliations:** 1Department of Chemistry and High Technology, Kuban State University, 350040 Krasnodar, Russia; devterova8julia@gmail.com (J.D.); map_kompass@mail.ru (A.N.); panyushkin@chem.kubsu.ru (V.P.); 2Department of Physics and Technics, Kuban State University, 350040 Krasnodar, Russia; kirillka.kir@bk.ru (K.K.); sokolovme@mail.ru (M.S.); 3Institute of Biochemical Technology, Ecology and Pharmacy, V.I. Vernadsky Crimean Federal University, 295007 Simferopol, Russia; shulvic@gmail.com (V.S.); galex0330@gmail.com (A.G.); 4Institute of Applied Synthetic Chemistry, Vienna University of Technology, A-1060 Vienna, Austria

**Keywords:** complex compound, terbium, 4-stearoylpyrazol-5-one, Langmuir film, Langmuir-Blodgett film, luminescence, IR microscopy

## Abstract

In this study, we investigated the effect of terbium ions (Tb^3+)^ on the subphases of the limiting area of the molecule for the complex compound (CC) TbL_3_∙bipy (where HL is 3-methyl-1-phenyl-4-stearoylpyrazol-5-one and bipy is 2,2′-bipyridine). We examined the Langmuir monolayer and the change in the luminescence properties of TbL_3_∙bipy-based Langmuir-Blodgett films (LBFs). The analysis of the compression isotherms, infrared, and luminescence spectra of TbL_3_∙bipy LBFs was performed by varying the concentration of Tb^3+^ in the subphases. Our results demonstrate the partial dissociation of the CC at concentrations of C(Tb^3+^) < 5 × 10^−4^ M.

## 1. Introduction

Lanthanide complex compounds in the form of ordered Langmuir-Blodgett molecular assemblies are interesting objects of research due to their various important practical properties (photophysical, spectral, magnetic, etc.) [1,2,3]. They are promising molecular structures in the development of hybrid materials for optics, integrated optoelectronics, photonics, sensorics, etc.

An important condition for obtaining highly luminescent LBFs based on Ln(III) (Eu^3+^, Tb^3+^, Sm^3+^, Dy^3+^, etc.) CCs is the efficient transfer of excitation energy from the triplet levels of ligands to the resonance levels of Ln(III) ions [4,5]. This condition is satisfied if the energy of the ligand triplet level is somewhat higher than the energy of the lanthanide resonance level (the optimal energy difference is 2500–3500 cm^−1^ for Eu(III) and 2500–4000 cm^−1^ for Tb(III)).

Additionally, to obtain LBFs, surface-active properties of the compounds used are necessary. This requirement is met by ligands with a hydrophobic hydrocarbon radical chain length from C_12_ to C_18_ [6]. However, the number of ligands that meets both requirements is extremely limited. Therefore, when obtaining luminescent LBFs based on CCs, several basic approaches are generally used.

In the first approach, surface-active ligands are initially used, which coordinate metal ions from the subphase by the process of complexation or ligand exchange at the air/water interface [7,8,9,10]. In this case, both classical surfactants (for example, fatty acids) and specially synthesized surfactants with coordinating groups can provide efficient transfer of excitation energy to the lanthanide ion (for example, β-diketones, 1,10-phenanthroline, 2,2′-bipyridine, etc.) The disadvantage of this approach is the difficulty of controlling the composition and structure of the CC formed as a result of the complexation of metal ions with a surfactant film, due to the complexity of equilibrium processes.

Another approach uses a mixture of nonsurface-active Ln^3+^ complexes and classical film-forming substances (for example, stearic acid or arachidic acid) to obtain luminescent LBFs [11,12,13,14,15]. In this case, the CC (for example, Ln(L_1_)_m_(L_2_)_n_, where L_1_ is a low-molecular-weight β-diketone and L_2_ is H_2_O, 1,10-phenanthroline, 2,2′-bipyridine, etc.) is poorly soluble in water, and its ligands are low-molecular-weight compounds without surfactant properties. As a result of this method, a pseudo-LBF is formed with luminescent centers that are inhomogeneous in size and distribution through the structure of the layer.

The third approach for obtaining luminescent LBFs is associated with the use of pre-synthesized complexes of ligands that meet both of the previously described requirements. The composition of this CC can be described as Ln(L_1_)_m_(L_2_)_n_, where either L_1_ or L_2_ can act as a ligand with surface-active properties. In the first case, L_1_ is a surfactant, and L_2_ is a low-molecular-weight ligand, such as H_2_O, 1,10-phenanthroline, 2,2′-bipyridine, etc. [16]. In the second case, L_2_ is a surfactant that binds the Ln(III) ion due to the formation of a π-complex, and L_1_ is a low-molecular-weight chelating ligand (for example, β-diketone) [17,18]. A variant of this approach is the use of an anionic CC of the composition [Ln(L_1_)_m_]^−^, where L_1_^−^ is a surfactant [16], or [Ln(L_1_)_m_]^−^(L_2_^+^)_n_, where L_2_^+^ is a cationic surfactant and L_1_ is a low-molecular-weight chelating ligand [19].

These described approaches make it possible to obtain luminescent LBFs based on Ln(III) complexes. However, their intensity of luminescence is much lower than that of the initial complexes, which is not suitable for practical application. We suggest this is due to the neglect of another important factor in the formation of LBFs based on CCs: the equilibrium of the complexation process and its establishment at the phase interface (aqueous medium/CC Langmuir film) between the CC in the Langmuir film and the initial components of the CC in the water subphase:$$M^{n+}+mL^{-}\;↔\;M^{n-m}L_{m}$$

In most studies to obtain LBFs based on CCs, including luminescent LBFs, pure deionized water was used as the subphase. The subphase pH also affects the complexation equilibrium. As a result, it shifts toward the destruction of CCs, and their hydrolysis (dissociation) is observed. Thus, the LBF consisting of the destroyed complexes and the free ligand is transferred to the substrate. This assumption was confirmed by our studies [20,21,22,23]. In this regard, when obtaining LBFs based on CCs, it is necessary to pay special attention to the effect of the composition of the subphase (concentration of the complexing agent ion, pH, etc.) on the important practical properties of the formed LBFs, as well as identifying the regularities of this effect.

Previously, we obtained a series of Tb(III) CCs with the diphilic 4-acylpyrazole-5-one [16]. These compounds possess unique photophysical properties (high quantum yields of luminescence, narrow-band emission, etc.). The production of LBFs based on these CCs on aqueous subphases with pH ~5.8 was described, and their luminescence spectra were studied. However, the intensity of the emission spectra was not high enough for practical application.

In the present study, we used a TbL_3_∙bipy CC (where HL is 3-methyl-1-phenyl-4-stearoylpyrazol-5-one (Figure 1) and bipy is 2,2′-bipyridine) for LBF formation. Compression isotherms on subphases with different concentrations of Tb(III) ions were studied, as well as changes in the molecular structure of ten-layer LBFs based on TbL_3_∙bipy using IR microscopy. Depending on the presence of Tb(III) in the subphase during LBF formation, their spectral-luminescence characteristics were also examined.

## 2. Materials and Methods

All starting reagents and chemicals were purchased from either Aldrich (München, Germany) or Merck (Darmstadt, Germany) and used without further purification. Solvents used for spectroscopic studies were purified and dried according to standard procedures before use. Organic ligands were obtained according to the literature methods.

π–A compression isotherms and LBF transfer were performed at a temperature of 24 °C on a KSV Minitrough II (KSV Instruments, Helsinki, Finland) equipped with a platinum Wilhelmy plate. TbL_3_∙bipy compression isotherms were recorded on aqueous subphases containing different concentrations of Tb(III) ions (1×10^−3^, 5×10^−4^, 1×10^−4^, 1×10^−5^, 1×10^−6^, and 1×10^−7^ M). The molecular area (A_0_) of the monolayer in the region of the crystalline phase was determined by extrapolation of the rectilinear section of the compression isotherm to a surface pressure of π = 0 mN/m. The measurement error was ±1 Å^2^. LBFs were formed by the vertical transfer of monolayers from the surface of the aqueous subphase onto glass plates (to study the spectral-luminescent characteristics of LBFs) with a 60 nm gold layer on one side and a 40 nm Cr sublayer (to record the IR spectra of LBFs). The transfer rate was 3 mm/min, and the pH of the subphase was 5.8, C(Tb^3+^) = 5×10^−4^ M. The average transfer rate for glass was 0.83 and 0.92 mm/min for glass with a gold layer. Single- and ten-layer TbL_3_∙bipy LBFs were obtained. The orientation of the layers in the obtained LBFs corresponded to the Y-type.

### 2.1. Preparation of Solutions

All solutions for the subphases with a given C(Tb^3+^) and pH were prepared by dilution with an appropriate volume of 0.01 M TbCl_3_ solution using deionized distilled water with a resistivity of 18 MΩ/cm. The initial 0.01 M terbium chloride solution was prepared by dissolving a sample of TbCl_3_ in deionized distilled water. Tb(III) content was determined by complexometric titration with Trilon B using xylenol orange as an indicator. pH of the solutions was adjusted to 5.8 by adding a concentrated solution of freshly prepared NaOH.

### 2.2. IR Spectroscopy and IR Microscopy

The IR spectra of TbL_3_∙bipy and HL were recorded in the 4000–600 cm^−1^ region on a Vertex 70 FT-IR spectrometer (Bruker, Ettlingen, Germany) in NPVO mode. Before recording the main spectrum, the baseline correction was used, and a preliminary spectrum of the compound was obtained. Next, software processing of the spectrum was carried out, and normalization and smoothing were used. The limit of acceptable absolute error of measurements on the wave number scale was ±0.05 cm^−1^.

The IR spectra of LBFs were recorded on a Nurex 2000 IR microscope (Bruker, Germany) combined with a Vertex 70 IR Fourier spectrometer in the 4000–600 cm^−1^ range with the reflection mode using the grazing angle objective (GAO) [24]. The GAO allowed recording the IR spectra of extremely thin coatings on the metallic surfaces by using the grazing angle incidence reflection technique. The IR spectra of ten-layer LBFs were recorded in several stages. First, the background spectrum of the substrate surface covered with the gold layer was recorded without an LBF. Then, a search for a layer of the LBF was carried out in the visual viewing mode of the image, with the aim of calibration at the maximum amplitude of the reflected signal intensity. Registration parameters (polarizer angle, number of accumulations, and resolution) were determined empirically. A high-quality optical image of the surface was captured by adjusting the image contrast and aperture diaphragm. To obtain the maximum signal intensity, the angle of rotation of the polarizers was adjusted. The number of accumulations for obtaining the resolved IR spectra of ten-layer LBFs was 3000 scans. The polarization angle was 45° with respect to the direction of LBF transfer to the substrate. The resolution was 4 cm^−1^. The final IR spectrum of the ten-layer film was obtained by subtracting the background spectrum from the accumulated one. A baseline correction was then used to normalize the intensity of the peaks after subtraction. After that, the spectra were smoothed to eliminate parasitic noise.

### 2.3. Luminescence Spectroscopy

Luminescence excitation spectra were recorded on a Fluorat-02-Panorama spectrofluorometer (Lumex, Saint Petersburg, Russia) with the following registration parameters: temperature, 296 K; excitation wavelength, from 200 to 450 nm; registration wavelength, 545 nm; averaging, 25 flashes; strobe duration, 2000 µs; and strobe delay, 10 µs.

Luminescence spectra of the initial complexes and LBFs at base level were recorded in the wavelength range of 430–700 nm using an AvaSpec-ULS2048L-USB2 high-sensitivity wave spectrometer (Avantes, Apeldoorn, The Netherlands) at 296 K. Excitation of luminescence was performed by means of an AvaLight-LED spectrometer module equipped with a UV-LED light with a wavelength maximum of 355 nm. The emission accumulation time for the initial TbL_3_∙bipy was 0.4 s with 500 accumulations and 10 s with 100 times for LBFs. 

Luminescence spectra measurements were carried out from 293 to 80 K using a Janis VPF-100 optical vacuum cryostat (Janis Research, Woburn, MA, USA) equipped with a Lakeshore Model 335 temperature controller (Lake Shore Cryotronics, Inc., Westerville, OH, USA).

## 3. Results

### 3.1. General Characterisation

TbL_3_∙bipy monolayers were characterized by good compressibility and a pronounced crystalline phase, which was seen in their compression isotherms (Figure 1). However, the minimum area per molecule in the condensed state varied upon variation of the concentration of Tb(III) in the subphases. The values of A_0_ determined for TbL_3_∙bipy on subphases with given values of Tb(III) concentrations are shown in Table 1. We found that, even at low concentrations of terbium cations (C = 1·10^−7^ M), the area per molecule in the TbL_3_∙bipy monolayer was much higher than the values obtained for the monolayer on the aqueous subphase not containing Tb(III) ions. The maximum value of A_0_ ≈ 120 Å^2^/mol was reached at C(Tb^3+^) = 1·10^−5^ M and did not change much with further increase in Tb(III) concentration (Figure 2). Such changes in A_0_ can be explained by the formation at concentrations of C(Tb^3+^) ≥ 1·10^−5^ M in the subphase equilibrium at the phase interface (aqueous medium/LF), which provided the highest monolayer content of nondissociated TbL_3_∙bipy. It should be noted that when LBF was obtained on the subphase in the presence of terbium ions, TbL_3_∙bipy transfer ratios were 0.8–0.9, while in the absence of Tb(III), they were 1.1–1.2. This indicated an increase in the film-forming properties of CCs and provided 10 layers of LBF.

### 3.2. IR Spectroscopy

In order to determine the reasons associated with the changes in A_0_ during the formation of LBFs, we studied the IR spectra of the initial compounds (TbL_3_∙bipy and HL) and the ten-layer LBFs obtained on subphases without terbium ions at C(Tb^3+^) = 5×10^−4^ M (Figure 3).

In the IR spectra of both the parent compounds and the LBFs based on them, intense absorption bands of the valence vibrations of C–H bonds were observed in the region of 2988–2980 cm^−1^, indicating the transfer of the CC on the substrate surface regardless of the presence of Tb(III) ions (Appendix A). As shown, the broad absorption band in the region of 1554 cm^−1^, resulting from the vibrations of conjugated C=C and C=O bonds of the ligand form, disappeared in the IR spectrum when HL was coordinated. Additionally, a number of intense absorption bands appeared in the regions of 1610–1620, 1500, and 1365–1370 cm^−1^ in the IR spectrum of TbL_3_∙bipy. According to data from the literature, these bands correspond to different types of valence vibrations of the coordinated β-dicarbonyl group [4]. The coordination of 2,2′-bipyridine is evidenced by the presence of C=C and C=N bond vibrations in the region of 1525–1580 cm^−1^ in the IR spectrum of the CC.

Regarding LBF, the main absorption bands attributed to the vibrations of the coordinated HL were observed in the IR spectra. However, in the case of LBF obtained on the subphase not containing Tb(III), there were no absorption bands of the coordinated 2,2′-bipyridine in the IR spectrum. This indicated that the formation of the LBF on the subphase surface in the absence of the complexing agent ions led to the detaching of the neutral ligand or its replacement by solvent molecules due to the shift in the equilibrium of the complexation process at the phase interface (aqueous medium/CC Langmuir film). This may result in a change in the geometry of the coordination site of the studied CC, which would lead to a change in the shape of the compression isotherm and, accordingly, to a decrease in A_0_.

### 3.3. Luminescent Properties

In order to investigate the effect of the presence of Tb(III) in the subphase on the luminescence properties of the obtained TbL_3_∙bipy-based LBFs, the excitation and emission spectra of the single- and ten-layer LBFs obtained on subphases containing no Tb(III) ions and at C(Tb^3+^) = 5 × 10^−4^ M were studied.

The excitation spectrum of the initial TbL_3_∙bipy (Figure 4) lay in the wavelength range of 200 to 450 nm and had several pronounced components. The most intense result had maximum excitation at a wavelength of 345 nm. However, for LBFs, the excitation spectra lay in a narrower range (200 to 350 nm) with a maximum wavelength of about 270 nm. This indicated the absence of intermolecular mechanisms of excitation energy transfer in the LBFs and the associated concentration quenching of luminescence. This result was confirmed by analyzing the changes in the excitation spectra of TbL_3_∙bipy luminescence in a chloroform solution at different concentrations. As can be seen (Figure 5, Figure 6 and Figure 7), with a decrease in the TbL_3_∙bipy concentration, the maximum of the excitation band shifted to the short-wavelength region. Simultaneously, an increase in intensity was observed.

The luminescence spectra of the initial TbL_3_∙bipy and its LBFs contained the emission bands that are characteristic of terbium(III) complexes corresponding with the transitions ^5^D_4_→^7^F_6_, ^5^D_4_→^7^F_5_, ^5^D_4_→^7^F_4_, and ^5^D_4_→^7^F_3_ (Figure 8). The splitting of the ^5^D_4_→^7^F_5_ transition band into three components indicated that TbL_3_∙bipy had a sufficiently high symmetry of the coordination node, which is described by a distorted square antiprism [25]. As shown, the luminescence intensity of terbium(III)-hypersensitive transitions was expected to increase with decreasing temperature, associated with the minimization of nonradiative excitation energy losses (a decrease in the internal and vibrational quenching of luminescence). Notably, the intensities of individual emission bands of hypersensitive transitions changed in different ways with decreasing temperature (Figure 8 and Figure 9). The largest changes in intensity relative to the initial values at 300 K were observed for bands with a λ_max_ = 541.7 nm for the hypersensitive transition ^5^D_4_→^7^F_5_ and λ_max_ = 490.9 nm for ^5^D_4_→^7^F_6_.

This was explained by the redistribution of the excitation energy between the channels of its transfer from the triplet levels of the ligands to the radiative sublevels of terbium(III) with a decrease in the vibrational mobility of the coordination bonds. In addition, upon a detailed examination of the line shape of the spectra of the hypersensitive transitions ^5^D_4_→^7^F_5_ and ^5^D_4_→^7^F_6_ (Figure 8), the appearance of new components as shoulders was observed with decreasing temperature. This may indicate the simultaneous presence of several nonequivalent forms of the coordination center with similar configurations of a strongly distorted square antiprism [26,27,28].

In the luminescence spectra of LBFs obtained on the subphase with C(Tb^3+^) = 5 × 10^−4^ M at room temperature (Figure 10), the shape of the line of the hypersensitive transition ^5^D_4_→^7^F_5_ was close to that of the luminescence spectra of the initial TbL_3_∙bipy at about 80 K. This may indicate an identical configuration of their coordination centers. Due to the limited sensitivity of the spectral equipment, the low resolution of other hypersensitive transitions did not allow them to be discussed in detail.

Regarding the LBF obtained on the subphase with C(Tb^3+^) = 0 M, the line shape of this hypersensitive transition had an intermediate value. The coordination center geometry was also a distorted antiprism. However, its nodes probably contained water molecules that replaced bipy due to hydrolysis (dissociation) of TbL_3_∙bipy on the aqueous subphase in the absence of Tb^3+^ ions. This assumption was confirmed by an analysis of the intensity of the LBFs’ luminescence spectra. Figure 10 shows that the luminescence intensity of the ten-layer LBFs obtained on a subphase with C(Tb^3+^) = 5 × 10^−4^ M was two times higher than that of LBFs obtained on subphases not containing Tb^3+^. It was also confirmed by the IR spectroscopy data on the detachment of the 2,2′-bipyridine molecule during the formation of TbL_3_∙bipy LBFs on subphases not containing Tb^3+^.

## 4. Conclusions

The analysis of the compression isotherms, infrared, and luminescence spectra of TbL_3_∙bipy LBFs obtained by varying the concentration of Tb(III) in the subphases demonstrated the partial dissociation of the CC at concentrations of C(Tb^3+^) < 5×10^−4^ M. In this case, the 2,2′-bipyridine molecule detached, which led to a significant decrease in the luminescence intensity of the obtained LBFs.

We suggest that this was due to the emergence of an equilibrium system at the phase interface (aqueous medium-Tb(III) complex-based Langmuir film), which was a result of the complexation reaction. The equilibrium in this system shifted toward the destruction of complexes due to inadequate pH for complex formation and an absence of Tb(III) cations in the aqueous medium.

## Data Availability

Data are presented in the article and Appendix A. The original data presented in this study are available on request from the corresponding author.

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
