# Peer review of "Features of the Preparation and Luminescence of Langmuir-Blodgett Films Based on the Tb(III) Complex with 3-Methyl-1-phenyl-4-stearoylpyrazol-5-one and 2,2′-Bipyridine"

_materials, 2022, doi:10.3390/ma15031127_

Round 1
Reviewer 1 Report
This manuscript described monolayer behaviors of a terbium complex on the subphase surfaces with various concentration of Tb3+ ions, and the luminescent behaviors of their LB films. Similar researches have been published in the past decades by several research groups, though not the same complex. However, it can be published, though not so interest. The authors provided a detailed description and explanation on their founding. Specific comments are as follows.
- Literature cited is too old, a lot of new publications are not included.
- Please use LB instead of Langmuir-Blodgett, except for the first use. This item has been well known.
- For the present complex, monolayer behaviors are mainly determined by the amphiphilic ligand, so the Tb3+ concentration has little effect on it.
- Excitation spectrum should be compared with the complex in dilute solution, not the powders; since in the latter case, aggregates have large effect on the luminescence.
- Concentration effect should be caused by the formation of aggregates or dimers or excimers when measured at higher concentrations of the complexes. Please refer to related literature.
- Fluorescence emission spectra were not reported. Whether they are hardly detected? If so, this may be related to the dimer formation, resulting in a lower triplet energy of the ligand.
Author Response
We are grateful to the reviewer for a careful analysis of the article and valuable comments. Please see below our responses to your comments
- Literature cited is too old, a lot of new publications are not included.
We agree with the comment. In accordance with your remark, some references have been replaced with newer ones.
- Please use LB instead of Langmuir-Blodgett, except for the first use. This item has been well known.
Corrections have been made as per your comment. The abbreviation “CC” has also been introduced instead of "complex compound".
- For the present complex, monolayer behaviors are mainly determined by the amphiphilic ligand, so the Tb3+ concentration has little effect on it.
Initially, we thought the same as you, but taking into account the fact that: 1) the air-water interphase, on which Langmuir films are formed, is a specific area of matter; 2) reversibility of the complexation process; 3) low luminescent characteristics of the formed LB films; we assumed the opposite. Our experimental data on the change in the area per molecule of the complex compound in the monolayer on the concentration of the terbium ion in the subphase (Figure 2) and the increase in the luminescence intensity of the LB films deposited in the presence of terbium ions (Figure 10) confirm this assumption.
- Excitation spectrum should be compared with the complex in dilute solution, not the powders; since in the latter case, aggregates have large effect on the luminescence.
We agree with your remark. However, the excitation spectra of solutions of the complex compound are presented separately in the text of the article (Figure 5-7). They are also discussed in comparison with the excitation spectra of the complex in the form of a powder and LB films based on them. If you consider it necessary to show the excitation spectra in one figure, then, in our opinion, in this case the figure will be overloaded.
- Concentration effect should be caused by the formation of aggregates or dimers or excimers when measured at higher concentrations of the complexes. Please refer to related literature.
Agree with your comment. We would like to draw your attention to the fact that in the text of the article we briefly discussed this without going into details. (Lines 220-221).
- Fluorescence emission spectra were not reported. Whether they are hardly detected? If so, this may be related to the dimer formation, resulting in a lower triplet energy of the ligand.
In this case, the triplet level of 1-phenyl-3-methyl-4-stearoylpyrazol-5-one was determined and described earlier in [1, 2]. It amounted to 21800 and 21980cm-1, respectively. This value corresponds to the efficient energy transfer to the excited 5D4 level of the terbium ion (ΔЕ(T1 → 5D4) = 1400 cm–1), whereas the gap between the Т1 and 5D0 levels of the europium ion is too large, being 4400 cm–1. Fluorescence of LB films based on the studied compound was not observed.
- Belousov Yu.A., Utochnikova V.V., Kuznetsov S.S., Andreev M.N., Dolzhenko V.D., and Drozdov A.A. New Rare-Earth Metal Acyl Pyrazolonates: Synthesis, Crystals Structures, and Luminescence. Properties Russian Journal of Coordination Chemistry, 2014, Vol. 40, No. 9, pp. 627–633.
- Shul’gin V., Pevzner N., Gusev A., Sokolov M., Panyushkin V., Devterova J., Kirillov K., Martynenko I., Linert W. Tb(III) complexes with 1-phenyl-3-methyl-4-stearoyl-pyrazol-5-one as a materials for luminescence Langmuir-Blodgett films. J. Coord. Chem., 2018, 71, 4228–4236. https://doi.org/10.1080/00958972.2018.1536783
Reviewer 2 Report
The manuscript entitled “Features of the preparation and luminescence of Langmuir-Blodgett films based 3 on the Tb(III) complex with 3-methyl-1-phenyl-4-stearylpyrazol-5-one and 4 2,2'-bipyridine” brings a description on the preparation of LB films containing a luminescent Tb(III) complex and the influence of the concentration of Tb(III) concentration in the subphase during the film deposition. I consider the manuscript is properly prepared, however some points are raised and listed below.
Line 34-35: Actually, if the energy of triplet is too high the transfer is not efficient and if it is equal to the resonant level of the Ln the sensitization would not be efficient either. There is an optimal energy of the triplet that depends on the ligand type and the lanthanoid.
Line 71: The Tb(III) comes from nowhere to the text. In the introduction, so far to this point, the authors are using the general lanthanide term, then it appears a Tb(III) without previous citation.
Line 73: this equation only makes sense if the ligand has a charge -1. Or authors could consider using ‘m HL’ on the left and H+ on the right to balance the equation.
Line 166: the word ‘that’ is duplicated.
Table 1, Figure 2: substitute the ‘,’ by ‘.’ For all numbers.
Figure 3: the inner legend of the lines has some missing characters.
Lines 194-197: the region between 2988-2980 cm^-1 is not shown in figure 3. Authors should represent the full range spectra in Figure 3 or represent the full range spectra in a Supplementary Material Section, which I believe is more appropriate.
Lines 187-189: Authors pose a question: What are the reasons for the change in A0? And they propose to answer by using IR. However, at the end of this topic, this question remained unclear.
Line 219: It is not seen in Figure 4 the most prominent component of Tb-complex at 345 nm. It is probably over 350 nm.
Author Response
We are grateful to the reviewer for a careful analysis of the article and valuable comments. Please see below our responses to your comments
- Line 34-35: Actually, if the energy of triplet is too high the transfer is not efficient and if it is equal to the resonant level of the Ln the sensitization would not be efficient either. There is an optimal energy of the triplet that depends on the ligand type and the lanthanoid.
In accordance with your remark, corrections have been made to the appropriate place in the publication as follows: "This condition is satisfied if the energy of the ligand triplet level is somewhat higher than the energy of the lanthanide resonance level (the optimal energy difference equals 2500-3500 cm-1 for Eu(III) и 2500-4000 cm-1 for Tb(III))."
- Line 71: The Tb(III) comes from nowhere to the text. In the introduction, so far to this point, the authors are using the general lanthanide term, then it appears a Tb(III) without previous citation.
In accordance with your remark, corrections have been made to the appropriate place in the publication.
- Line 73: this equation only makes sense if the ligand has a charge -1. Or authors could consider using ‘m HL’ on the left and H+ on the right to balance the equation.
In accordance with your remark, corrections have been made to the appropriate place in the publication by adding a charge -1 to the ligand.
- Line 166: the word ‘that’ is duplicated.
In accordance with your remark, corrections have been made to the appropriate place in the publication.
- Table 1, Figure 2: substitute the ‘,’ by ‘.’ For all numbers.
In accordance with your remark, corrections have been made to the appropriate place in the publication.
- Figure 3: the inner legend of the lines has some missing characters.
In accordance with your remark, corrections have been made to the appropriate place in the publication.
- Lines 194-197: the region between 2988-2980 cm^-1 is not shown in figure 3. Authors should represent the full range spectra in Figure 3 or represent the full range spectra in a Supplementary Material Section, which I believe is more appropriate.
In accordance with your remark, the full range IR spectra of the studied compounds are presented in Supplementary Materials.
- Lines 187-189: Authors pose a question: What are the reasons for the change in A0? And they propose to answer by using IR. However, at the end of this topic, this question remained unclear.
In accordance with your comment, the following text has been added to the IR spectroscopy section: " This indicates that the formation of the LBF on the subphase surface in the absence of the complexing agent ions leads to the detaching of the neutral ligand or its replacement by solvent molecules due to the shift in the equilibrium of the complexation process at the phase interface (aqueous medium / CC Langmuir film). This may result in a change in the geometry of the coordination site of the studied CC, which leads to a change in the shape of the compression isotherm and, accordingly, to a decrease in A0."
- Line 219: It is not seen in Figure 4 the most prominent component of Tb-complex at 345 nm. It is probably over 350 nm.
A detailed examination of Figure 4 shows that the maximum of the emission band of the complex is at a wavelength of 346 nm.

Round 2
Reviewer 1 Report
The manuscript can be accepted for publication.